# OmniCam: Unified Multimodal Video Generation via Camera Control

## Abstract

Camera control, which achieves diverse visual effects by changing camera position and pose, has attracted widespread attention. However, existing methods face challenges such as mono-interaction and scene generalization. To address these issues, we present OmniCam, a unified multimodal camera control framework. Leveraging large language models and video diffusion models, OmniCam generates spatio-temporally consistent videos. It supports various combinations of input modalities: the user can provide text or video with expected trajectory as camera path guidance, and image or video as content reference, enabling precise control over camera motion. To facilitate the training of OmniCam, we introduce the OmniTr dataset, which contains a large collection of high-quality long-sequence trajectories, videos, and corresponding descriptions with totally 10K+ scenes. Experimental results demonstrate that our model achieves state-of-the-art performance in high-quality camera-controlled multimodal video generation across various metrics. DemoPage can be found in `https://flexcam.github.io/`.

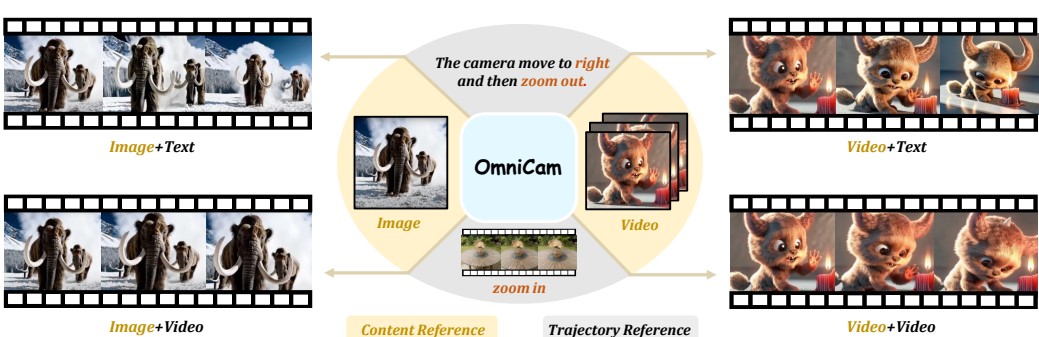

Figure 1: **An overview of OmniCam.** Given diverse modalities of content references and trajectory guidance, OmniCam generates high-quality video sequences by camera motion control. Specifically, OmniCam integrates various combinations of content (e.g., image or video) and trajectory (e.g., text instructions or camera motion from video) references. This approach allows OmniCam to accurately synthesize videos consistent with user-specified inputs.

## 1 Introduction

Camera control refers to the task of expressing different visual effects by controlling camera position and pose. The essence of camera movement is a parameterized representation of the four-dimensional space-time continuum, a recognition that is becoming an important breakthrough in spatial intelligence research.

Current development in this field faces several challenges: I. Current methods have high interaction costs and support limited modalities. a) Current methods lack exploration of guiding trajectory generation using the text modality. However, text-guided trajectories have many advantages; for example, they are easy for users to interact with, which makes large-scale annotation possible. Additionally, since text is a modality supported by most I2V models, it can be used for pretraining foundational models. b) In video-to-video trajectory transfer, consistency in the magnitude of

camera motion is often poorly maintained. This is because methods based on external parameter estimation are sensitive to different scales in scenes. II. There is a lack of sufficiently versatile datasets to support these tasks. Therefore, we aim to build a viewpoint generation method based on discrete motion representation that supports long control sequences and multimodal control, as well as to create a sufficiently long and diverse dataset to facilitate this task.

To this end, we propose OmniCam, which initializes videos following target trajectories, then inpaints videos using a video diffusion model. It supports frame-level control by setting the start and end frames of each operations; supports compound movements in any direction, camera push-pull, allowing movement and rotation to any degree; supports speed control, providing a foundation for quick cuts; supports seamless connection of multiple operations, supporting long sequence operations, allowing continuous execution of multiple instructions; and allows common special effects such as camera rotation. Additionally, our model supports multimodal inputs, consisting of two parts: one providing content information and one providing trajectory information. The content part can be an image or video, and the trajectory part can be provided in two ways: through text descriptions of how the camera moves, or through a video with camera movement effects, from which OmniCam extracts the discrete motion representation using a designed VLM and applies it to the target video.

To train the OmniCam model, we present the OmniTr dataset, the first multimodal camera control dataset, including a large number of multi-stage trajectories and their corresponding videos and text descriptions. Each description includes multiple sub-instructions, with each sub-instruction recording information such as start time, end time, speed, direction, and rotation. Compared to other datasets, our dataset provides a large number of scenes with rich and significant camera movements.

We conducted quantitative and qualitative experiments, demonstrating that OmniCam can achieve flexible and complex trajectory control through multiple modalities. Our contributions are summarized as follows:

- **A perceptually aligned unified multimodal trajectory representation:** the discrete motion representation serves as a bridge across different modalities, addressing the inconsistency between perceived camera motion speed and absolute camera motion speed caused by scenes of varying scales.

- **A benchmark for video camera motion style transfer and text-guided camera control:** the OmniCam model is the first to support both dynamic and static states for video camera motion transfer as well as text-guided camera control.

- **A multimodal data engine for text and video:** the OmniTr dataset contains a large number of videos with identical camera motion trajectories and corresponding fine-grained annotations, which can be used for video-to-video camera motion transfer and training of text-guided camera control.

## 2 RELATED WORK

### 2.1 VIDEO DIFFUSION MODELS

Recent work on large-scale video diffusion models has achieved high-quality video generation. Video Diffusion Model Ho et al. (2022b) employs a 3D UNet architecture to jointly learn from images and videos. Imagen Video Ho et al. (2022a) introduces a cascade structure consisting of seven diffusion models, ingeniously combining key components such as TSR Yi et al. (2019); Xiang et al. (2020) and SSR Ledig et al. (2017); Wang et al. (2019) for efficient video generation. With the remarkable image quality achieved by text-to-image (T2I) generation models like Stable Diffusion Rombach et al. (2022b), numerous recent works focus on extending pre-trained T2I models by incorporating temporal modules. Align your latents Blattmann et al. (2023b) proposes a noise map alignment approach, effectively transforming image generation models into video generators. AnimateDiff Guo et al. (2023) opts to inject temporal modules into fixed spatial feature layers, creating a framework that enables personalized animation creation without fine-tuning. To enhance temporal consistency, Lumiere Bar-Tal et al. (2024) replaces conventional temporal super-resolution modules, directly generating full-frame-rate videos. Recently, researchers have increasingly introduced Transformer Vaswani (2017) architectures into video generation. SORA Brooks et al. (2024) has

**Trajectory Description**    First, the camera moves quickly downward. 1 to 2 seconds, it slowly pans to the right. Immediately after, the camera swiftly moves upward-right at a 30-degree angle. 3 to 4 seconds, it slowly pulls back.

*Discrete Motion Representation*

*<st>0</st><sep><ed>1</ed><sep><speed>high</speed><sep><d>down</d><sep><r>stay</r><sep><st>1</st><sep><ed>2</ed><sep><speed>slow</speed><sep><d>right</d><sep><st>2</st><sep><ed>3</ed>...<d>upright30</d>... <speed>slow...<d>backward</d>...*

*Trajectory*

*(0.0,0.0,0.0),(0.0,1.875,0.0),(0.0,3.75,0.0),(0.0,5.625,0.0),...,(0.0,15.0,0.0),(12.5,15.0,0.0),...,(15.0,15.0,0.0),...,(15.0,15.0,0.2),(15.0,15.0,0.4)...*

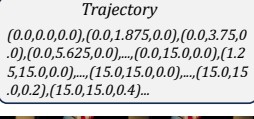

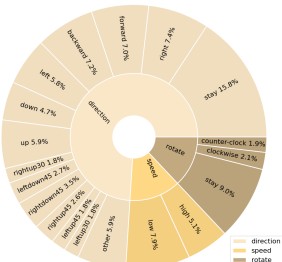

Figure 2: **OmniTr dataset consists of four key components:** trajectory description, discrete motion representation, trajectory, and corresponding video sequence. Notably, we visualized the discrete motion representations, with the pie chart on the right clearly showing the distribution proportions of various motion attributes. As can be seen, our dataset comprehensively covers all motion attributes.

made significant progress in generating realistic long videos utilizing the DiT Peebles & Xie (2023) architecture. Latte Ma et al. (2024) experiments with four DiT variants for spatiotemporal modeling in latent space, ultimately achieving coherent and realistic generation results.

## 2.2 CONTROLLABLE VIDEO GENERATION

With the rapid development of generative models across different input modalities, providing diverse guidance for precise control has become a research focus. Recent works such as SparseCtrl Guo et al. (2025) and SVD Blattmann et al. (2023a) utilize images as control signals for video generation. Because of the significance of camera motions in videos, camera control has gained increasing attention. AnimateDiff Guo et al. (2023) employs efficient LoRA Hu et al. (2021) fine-tuning to obtain model weights for specific shot types. Direct-a-Video Yang et al. (2024a) introduces a camera embedder to control camera poses during video generation; however, with three camera parameters, the model supports only basic camera controls like leftward panning. While Runway Research (2023) allows users to freely set camera movements, it is primarily limited to lens-centric operations and suffers from constraints in both movement magnitude and frequency. GameCrafter Li et al. (2025a) and Yan Fu et al. (2025a) translate camera control into keyboard-mouse operations for gaming, enabling precise camera state manipulation through gaming control concepts. However, this approach relies heavily on keyboard-mouse inputs, making it difficult to achieve camera control via simple natural language. TrajectoryCrafter Mark et al. (2025) attempts to implement text-based camera control while ensuring that the generated video does not deviate significantly from the original video through a Dual stream Conditional control, but it lacks fine-grained control. Unlike them, OmniCam simultaneously achieves text-based modal camera control and fine-grained control.

Regarding trajectory extraction from video modality, although traditional pose estimation algorithms have explored this field, they perform poorly in continuous camera trajectory estimation, particularly in low-frame-rate scenarios. This limitation stems from two technical bottlenecks: First, traditional methods rely on feature point matching algorithms like SIFT Lowe (2004), requiring sufficient visual overlap between adjacent frames. In low-frame-rate scenarios, large camera movements often lead to feature-matching failures. Second, increased inter-frame intervals significantly affect the prediction accuracy of motion models, increasing uncertainty in motion estimation. Our approach not only supports text-based guidance but also effectively extracts camera trajectories from target videos in low-frame-rate environments, providing a more flexible and robust solution for controllable video generation.

## 3 OMNITR DATASET

Existing datasets lack complex, flexible camera movements and multimodal input. To address these issues, we introduce the OmniTr dataset, a large-scale resource library specifically designed for comprehensive camera control.

As shown in Fig. 2, OmniTr uses trajectory groups as its basic unit. Each group contains four components: trajectory description, discrete motion representation, trajectory in polar coordinates, and high-quality videos. We carefully constructed 1K unique trajectory groups, generating a com-

Table 1: **Comparison of other datasets with OmniTr.** None of the other datasets include textual descriptions of the trajectories. T in the table stands for Text.

| Datasets | Modality | Any-direction | Zoom | Rotate | Speed | Complex |
|---|---|---|---|---|---|---|
| Tanks&Temples | Video | ✓ | ✓ | ✗ | ✗ | ✓ |
| RealEstate10k | Video | ✓ | ✓ | ✓ | ✗ | ✓ |
| CO3D | Image | ✗ | ✗ | ✗ | ✗ | ✗ |
| Webvid10m | Video | ✗ | ✗ | ✗ | ✗ | ✓ |
| ReCamMaster | Video | ✓ | ✓ | ✓ | ✗ | ✓ |
| OmniTr | T+V | ✓ | ✓ | ✓ | ✓ | ✓ |

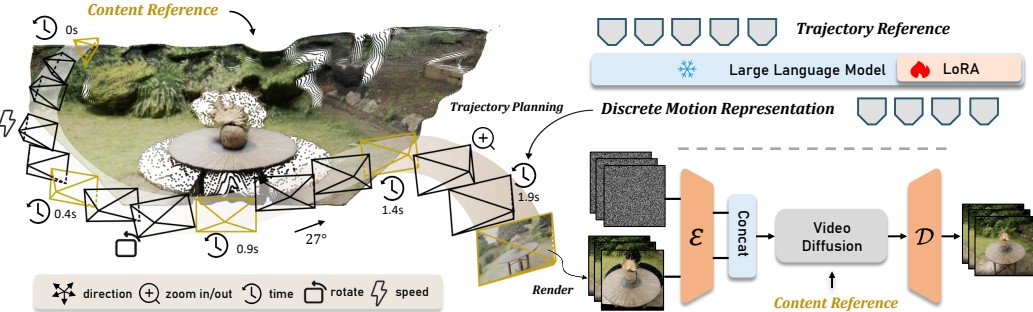

Figure 3: **An overview of OmniCam model pipeline.** After receiving the trajectory reference, OmniCam first converts it into discrete motion representations through large model. Subsequently, OmniCam uses a trajectory planning algorithm to calculate the camera pose for each frame based on these motions. Combined with the content reference, OmniCam renders the initial view for each frame. Finally, it utilizes a diffusion model to complete unknown regions in the new viewpoints, and stitch all frames together to generate a coherent video.

prehensive camera control dataset containing 1K trajectories, 10K descriptions, 30K videos, and their corresponding discrete motion representations. Among them, each trajectory corresponds to 10 descriptions and 30 videos of common categories.

OmniTr builds its video content based on the CO3D Reizenstein et al. (2021) dataset and uses large language models to generate diverse textual descriptions. The dataset provides control information accurate to the frame level, with discrete motion representations that can be directly converted into complete six-degrees-of-freedom (6DoF) sequences, enabling precise camera trajectory control.

As shown in Tab. 1, we compare the OmniTr dataset with existing datasets. Our dataset covers diverse camera movement trajectories and rich scene variations. Each text description consists of 1 to 5 camera operation descriptions, with each operation associated with a specific discrete motion representation. The text precisely expresses the time range, speed, direction, and angle of operations. The discrete motion representation contains several key fields: **starttime** and **endtime** describes the operation time period; **speed** describes the camera movement speed (low indicates slow, high indicates fast); **direction** describes the movement direction (including basic directions and combination directions at any angle); **rotate** describes the rotation method (clockwise, counterclockwise, or stationary). The pie chart in Fig. 2 demonstrates the dataset's comprehensive coverage of various operation methods.

To enhance the dataset's utility, we improve the text quality in multiple dimensions. **Time robustness processing**: Some operations explicitly specify a time range (such as "0 to 1 second"), while for others without explicitly specified times, default values are used: camera movement defaults to 1 second, and camera rotation defaults to 0.5 seconds. Operation times in some texts are non-continuous; for example, the first operation occurs at 0-1 seconds, while the second operation jumps to 3-4 seconds, with no additional operations during the intermediate time; **Angle flexibility processing**: The input text allows for any angle in combination directions; **Language styles diversifying**: We include formal statements, concise expressions, exaggerated descriptions, and other forms to ensure rich semantics and adaptability to different scenario requirements.

# 4 METHOD

## 4.1 TRAJECTORY GENERATION

Our camera trajectory generation system aims to produce appropriate trajectories based on input descriptions. Our method achieves frame-level precision through fine-grained control, supporting compound movements in arbitrary directions and camera zoom operations. Furthermore, our approach enables seamless integration of multiple operations, resulting in more natural and fluid camera trajectories. The system accommodates multi-modal inputs, accepting both text and video sequences as trajectory references.

### 4.1.1 DESCRIPTION TO DISCRETE MOTION REPRESENTATION

We utilize discrete motion representation as an intermediate representation for generating continuous trajectories, facilitating long-range fine-grained control. For textual inputs, a large language model is used to transform description into discrete motion representation, comprising sequences of <starttime, endtime, speed, direction, rotate>.

We fine-tune the large language model using LoRA (Low-Rank Adaptation). The input sequence can be represented as $Y = \{y_1, y_2, ..., y_k\}$. The discrete motion representation comprises multiple primitive statements, each containing a quintuple of control signals and separated by $_{<sep>}$ tokens. The resulting discrete motion representation sequence is formulated as $\hat{Y} = \{\hat{y_1}, \hat{y_2}, ..., \hat{y_t}\}$, where $t \in R^N$. Here, $N$ denotes the number of control signals derived from the description and satisfies the condition $5|N$. The loss function for discrete motion representation prediction is defined as:

$$\mathcal{L}_{\text{trajectory}} = -\sum_{t=1}^{T} \log p(y_t | \hat{y}_{<t}) \tag{1}$$

where $T$ denotes the total length of the trajectory sequence. $p(y_t|\hat{y}_{<t})$ represents the probability of correct model prediction and $\hat{y}_{<t}$ indicates all predicted results prior to time step $t$.

Similarly, video-guided camera control also utilizes discrete motion representation as a bridge. To address the issue of inconsistent motion magnitude across scales during transfer, we fine-tuned a vision-language model (VLM). Specifically, we divided the scenes in the dataset into three subsets, each containing scenes of similar scale. We set smaller thresholds for small-scale scenes and larger thresholds for large-scale scenes, so that the camera motion magnitude aligns with subjective perception. Additionally, since datasets such as RealCam-Vid Zheng et al. (2025) are based on external parameter trajectories, but the estimation of dynamic object camera extrinsic parameters contains unestimable errors, we adopt discrete motion representation as a bridge. Leveraging its ease of annotation, we manually corrected obviously erroneous samples. For example, in a case of "camera follows a running car," because the subject occupies a large portion of the frame and is relatively static with respect to the camera, the extrinsic parameter estimation incorrectly showed a stationary camera, which required manual correction.

### 4.1.2 DISCRETE MOTION REPRESENTATION TO TRAJECTORY

After obtaining the discrete motion representation, we employ the **trajectory planning algorithm** to calculate the spatial position of each point in the trajectory. The spatial position is parameterized by a triplet $(\phi, \theta, r)$, which includes the azimuth angle $\phi$, polar angle $\theta$, and radius $r$. The rotation is independently controlled by the rotate parameter in the discrete motion representation. These positions are then converted into a sequence of camera extrinsic parameters. This process is lossless and reversible. This algorithm models the camera movement around the object's center as a spherical motion by default. Initially, we compute the start and end frames affected by the control signals. Subsequently, based on specific control information, we calculate an incremental change $\delta = F(v, d)$, where $v$ and $d$ represent the control information for velocity and direction, respectively. The function accumulates corresponding angular or distance changes $\Delta = \Sigma \delta_t$ for each frame to generate the complete trajectory sequence.

Specifically, given a motion instruction, we first calculate the number of frames based on the frame rate and duration. Let $f$ denote the frame rate and $t$ represent the total duration; the total number of frames is expressed as $f \times t$. Through this approach, utilizing the concept of division points in a given ratio Mezouar & Chaumette (2000), we unify temporal and frame-based control. To simulate velocity variations, we initialize a unit incremental change $\delta^v$ and apply scaling factors, supporting control granularity at both high and low levels. For translational operations, the pose of each frame is computed by adding the increment to the pose of the previous frame, thereby forming a complete pose sequence for the corresponding operation. Rotational operations are implemented according to the details in the supplementary.

### 4.2 TRAJECTORY-GUIDED VIDEO SYNTHESIS

#### 4.2.1 RECONSTRUCTION AND RENDER

For image $I_{\text{ref}}$ serving as content references, we first use the dense stereo model Wang et al. (2024a) to extract its point cloud data, camera intrinsic parameters, and camera pose $C_{\text{ref}}$. Next, navigate the camera using the camera pose sequence $C = \{C_0, \ldots, C_{L-1}\}$ (including $C_{\text{ref}}$), render the point cloud, and generate a series of rendering results $P = \{P_0, \ldots, P_{L-1}\}$. Our goal is to learn the conditional distribution $x \sim p(x|I_{\text{ref}}, P)$ and generate high-quality perspective conversion videos $x = \{x_0, \ldots, x_{L-1}\}$ by rendering point cloud $P$ and reference image $I_{\text{ref}}$.

For videos serving as content references, we use the dynamic point-cloud reconstruction Zhang et al. (2024) to extract its point cloud data. Then, the target trajectory is used to render a video with holes, which serves as preparation for the downstream inpainting task.

#### 4.2.2 RESOLVE THE UNKNOWN REGION

As shown in Fig. 3, point cloud rendering results typically contain unknown regions. Similar to how humans can imagine the back of an object based on its front view, pretrained diffusion models Yang et al. (2024b) also possess this imaginative capability based on prior knowledge, which is why we apply them to complete these unknown regions.

To improve computational efficiency, we employ a DiT Rombach et al. (2022a) architecture. Inspired by previous work Xing et al. (2025); Yu et al. (2024), we construct a high-quality paired dataset containing point cloud rendering sequences $P = \{P^0, \ldots, P^{L-1}\}$ and corresponding real reference images $I = \{I^0, \ldots, I^{L-1}\}$.

During training, we freeze the VAE Kingma (2013); Kingma et al. (2013) encoder-decoder parameters and focus on optimizing the latent space. Specifically, we first encode the training data $I$ and $P$ into latent variables $z = \{z^0, \ldots, z^{L-1}\}$ and condition signals $\hat{z} = \{\hat{z}^0, \ldots, \hat{z}^{L-1}\}$ respectively. Then, we add sampled noise $8\epsilon$ (where $\epsilon \sim \mathcal{N}(0, I)$, meaning $\epsilon$ follows a normal distribution with a mean of 0 and an identity matrix as the covariance matrix) to the latent variables $z$. Finally, we concatenate the noisy $z$ with the condition signals $\hat{z}$ along the channel dimension. The model is optimized using the following diffusion loss function:

$$\min_\theta = \mathbb{E}_{t \sim \mathcal{U}(0,1), \epsilon \sim \mathcal{N}(0,I)} \left[ \|\epsilon_\theta(z_t, t, \hat{z}, I_{\text{ref}}) - \epsilon\|^2 \right] \tag{2}$$

where $z_t = \alpha_t z_0 + \sigma_t \epsilon$. Additionally, we inject the CLIP Radford et al. (2021) features of the reference image as conditions into the DiT to prevent domain shift.

### 4.3 PERCEIVED SPEED ADAPTATION

Traditional pose estimation is very sensitive to scene scale. For example, the sensitivity to changes in extrinsic parameters differs between large and small scenes, which can cause problems when they are evaluated together. In a large scene, a numerical large extrinsic parameter change $\delta$ (specifically, $\phi$ change and $\theta$ change) might be interpreted as a small camera movement speed, whereas the same $\delta$ in a small scene would be considered a large speed. This is because, perceptually, the camera motion corresponding to a $\delta$ change in a large scene is indeed subtle, while the same $\delta$ in a small scene corresponds to a significant camera movement. Since we aim to obtain perceived speed rather than absolute speed, we detach the evaluation according to different scene scales. Specifically, for the combined dataset, we classify scenes into three buckets—large, medium, and small—using an

operator. For each bucket, we define scales of extrinsic parameter changes based on their distributions, enabling speed evaluation based on perception rather than relying solely on absolute extrinsic parameter changes.

## 5 EXPERIMENT

### 5.1 IMPLEMENT DETAILS

For scenarios where image-provided content references are used, training is conducted on RealEstate10K and OmniTr dataset; for scenarios where video-provided content references are used, training is performed on MiraData and a private dataset created with RoboTwin, which includes some cases with long sequences and large-magnitude viewpoint variations. For the VLM, we use Qwen2.5-VL as the backbone on the RealCam-Vid and private datasets. We train the model and variants on 8 NVIDIA A100 GPUs. More specific implementation details can be found in the supplementary.

Table 2: The performance of different modalities (including content reference and trajectory reference) under different backbones across multiple indicators. VLM: Qwen2.5-VL; LLM: Llama.

| Modality | Backbone | Trajectory | | | | | | Quality | Consist |
|---|---|---|---|---|---|---|---|---|---|
| | | $M_{d-course}$ | $M_{d-fine}$ | $M_{speed}$ | $M_{rotate}$ | $M_{starttime}$ | $M_{endtime}$ | NIQE ↓ | CLIPSR ↑ |
| Text | LLM-SFT | 83.153 | 74.362 | 78.133 | 78.133 | 78.133 | 78.133 | 2.831 | 0.925 |
| Text | VLM-SFT | 83.251 | 64.519 | 70.763 | 71.655 | 74.827 | 72.844 | 2.830 | 0.926 |
| Video | MonST3R | 64.114 | 6.529 | 24.575 | 46.929 | 53.145 | 49.758 | 2.831 | 0.924 |
| Video | VLM-SFT | 82.400 | 12.199 | 33.6 | 63.099 | 74.600 | 66.900 | 2.830 | 0.926 |

Table 3: The comparison with other models. More comparisons with other methods can be found at supplementary and demo page. Both videos and images, as content, have been tested.

| Method | RotErr↓ | TransErr↓ | LPIPS↓ | PSNR↑ | FID↓ |
|---|---|---|---|---|---|
| CameraCtrl He et al. (2024) | 6.42 | 5.79 | 0.29 | 18.37 | 69.40 |
| LucidDreamer Liang et al. (2024) | 7.99 | 10.85 | 0.40 | 14.13 | 71.43 |
| CamI2V Zheng et al. (2024) | 5.98 | 5.22 | 0.26 | 18.27 | 58.30 |
| ZeroNVS Sargent et al. (2024) | 8.56 | 10.31 | 0.43 | 14.24 | 105.8 |
| MotionCtrl Wang et al. (2024c) | 8.08 | 9.29 | 0.38 | 16.29 | 70.02 |
| ReCamMaster Bai et al. (2025) | 1.21 | 4.75 | 0.39 | 21.77 | 57.10 |
| OmniCam (Ours) | **1.15** | **2.63** | **0.17** | **22.11** | **34.31** |

### 5.2 METRICS

To evaluate the accuracy of the generated trajectories, we propose five metrics to supervise the discrete motion representation. $M_{starttime}$ and $M_{endtime}$ are used to evaluate the model's accuracy in determining the start and end times. $M_{speed}$ is employed to assess the model's understanding of speed. $M_{rotate}$ is utilized to evaluate whether the model correctly comprehends the direction of rotation. $M_{direction}$ is used to judge the model's ability to accurately understand the direction of camera movement. All these metrics are essentially accuracy measures, calculated as the average of all subtasks. Among these metrics, $M_{starttime}$ and $M_{endtime}$ are evaluated independently, while $M_{rotate}$, $M_{speed}$, and $M_{direction}$ are assessed based on the condition that both the start time and end time are correctly determined. Additionally, due to the complexity of directional information, we divide $M_{direction}$ into $M_{d-course}$ and $M_{d-fine}$. $M_{d-course}$ allows for discrepancies in degrees but requires the direction to be correct, while $M_{d-fine}$ demands both the direction and degrees to be accurate. Since the discrete motion representation uniquely determines the trajectory, these metrics can evaluate the accuracy of the model-generated trajectories. We are the first to propose evaluation criteria for extracting complex trajectories from text or video, thereby laying a foundation for future research.

Additionally, we evaluate the generated videos using several classic metrics, such as LPIPS Zhang et al. (2018), PSNR Tanchenko (2014), SSIM Hore & Ziou (2010); Nilsson & Akenine-Möller

(2020), FID Jayasumana et al. (2024), NIQE Mittal et al. (2012), and CLIPSR Hessel et al. (2022). These metrics assess the video quality and temporal smoothness, and more details can be found in the supplementary materials.

We then calculate the rotation distance (RotErr) in comparison to the ground truth rotation matrices of each generated novel view sequence, expressed as:

$$\text{RotErr} = \sum_{i=1}^{n} \arccos \frac{tr(\boldsymbol{r}_{\text{gen}}^i \cdot \boldsymbol{r}_{\text{gt}}^{iT}) - 1}{2}, \tag{3}$$

where $\boldsymbol{r}_{gt}^i$ and $\boldsymbol{r}_{gen}^i$ denote the ground truth rotation matrix and generated rotation matrix. We also compute the translation distance (TransErr), expressed as:

$$\text{TransErr} = \sum_{i=1}^{n} \|\boldsymbol{t}_{\text{gt}}^i - \boldsymbol{t}_{\text{gen}}^i\|_2, \tag{4}$$

where $\boldsymbol{t}_{gt}^i$ and $\boldsymbol{t}_{gen}^i$ denote the ground truth translation matrix and generated translation matrix.

From 0 to 1 seconds, the camera rapidly **descends**. From 1 to 3 seconds, the camera slowly moves to the **left**.

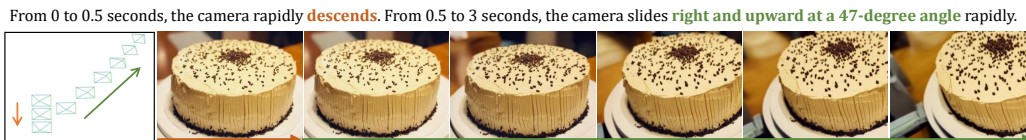

From 0 to 0.5 seconds, the camera rapidly **descends**. From 0.5 to 3 seconds, the camera slides **right and upward at a 47-degree angle** rapidly.

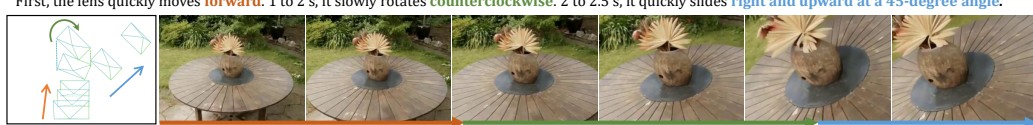

First, the lens quickly moves **forward**. 1 to 2 s, it slowly rotates **counterclockwise**. 2 to 2.5 s, it quickly slides **right and upward at a 45-degree angle**.

Figure 4: **Text description for camera control**. Each set of results demonstrates the generation effects of different types of camera motion combinations, including directional movements at specified angles, rotations, and other complex movements.

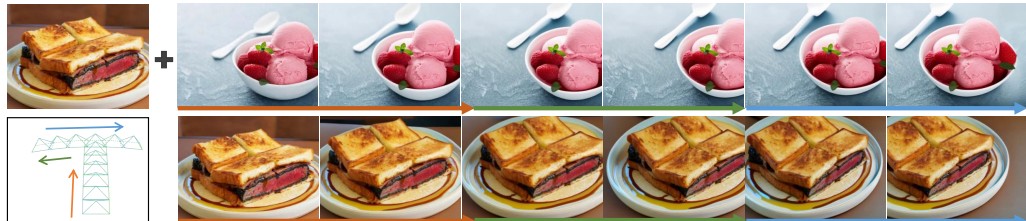

Figure 5: **Video trajectory for camera control**. OmniCam transfers the trajectory extracted from the input video to the output video. The first line represents the input and the second line represents the output.

## 5.3 MAIN RESULTS AND ABLATION STUDY

As shown in Tab. 2, for the task of obtaining trajectories through text descriptions, we compared two backbone models. The experimental results demonstrate that the LLM performs better. This is because the LLM is more focused on text comprehension, exhibiting stronger understanding capabilities compared to the VLM (Vision Language Model). For obtaining discrete motion representations from videos, we compared traditional camera pose estimation methods with training a

Table 4: Ablation on modules. DownSPL PCD refers to downsampling the point cloud to verify the impact of sparse point clouds on reconstruction.

| Model | RotErr↓ | TranErr↓ | PSNR↑ | FID↓ |
|---|---|---|---|---|
| Planner-only | 2.55 | 4.89 | 6.37 | 271.52 |
| Repaire-only | 3.15 | 13.43 | 23.11 | 26.30 |
| DownSPL PCD | 1.73 | 3.24 | 9.42 | 171.51 |
| OmniCam | 1.15 | 2.63 | 22.11 | 34.31 |

VLM model to directly estimate the discrete motion representation. The results show that VLM-SFT performs better, as in dynamic scenes—especially those where the subject occupies a large proportion—higher-level semantic understanding is required for accurate estimation. Additionally, this aligns with the requirements of perceptual speed evaluation. Further discussions can be found in the appendix.

Table 3 presents a quantitative comparison of the generation quality of several effective methods and their similarity to the real distribution. Additionally, comparison results with a broader range of models are available on the demo page and in the supplementary materials. The experimental results indicate that our method excels in generating higher-quality videos. As shown in Fig. 4 and Fig. 5, we visualize the effects of controlling the camera trajectory via text and via video.

As shown in Tab. 4, using only the Trajectory Planning as the final output results in a black background and missing region in the output. In contrast, using only the Repair module—conditioned on the input frame and discrete trajectory points instead of the rendered view—leads to larger rotation and translation errors. We downsample the point cloud by a factor of 1/8, with the results shown in Tab. 4. Information loss in sparse point clouds causes spatial distortions, which in turn degrade the quality.

## 5.4 GENERAL DISCUSSION AND HUMAN STUDY

We conduct a comparison of state-of-the-art methods across various domains, examining the potential of different technical routes and highlighting the shortcomings of reconstruction methods, 4D reconstruction, and other approaches. Given the significant functional differences among models in various fields, adopting a unified quantitative evaluation metric is neither fair nor feasible. Therefore, we employ a manual evaluation method, inviting 50 participants to rate the methods on a scale of 1 to 5, with the final scores rounded. For Boolean evaluations, such as whether or not it is open source, open source is rated 5 out of 5 and not open source is rated 1 out of 5. The results are shown in Fig. 9 in A.7. The experimental findings indicate that ViewCrafter Yu et al. (2024) is cumbersome to interact with, especially when handling complex instructions, and it does not support learning camera trajectories from videos. ZeroNVS Sargent et al. (2024) is a novel view synthesis algorithm, but it can only generate one frame at a time and is cumbersome to use. GenWrap Seo et al. (2024), another novel view generation algorithm, offers fast inference speed but suffers from generalization issues. CAT4D Wu et al. (2024), a 4D model, is hindered by its slow speed and lack of open-source availability. Additionally, reconstruction methods like One-2-3-45++ Liu et al. (2024) are designed for single-object reconstruction and do not include scenes, so they are not considered in our study.

## 6 CONCLUSION

OmniCam is an unified multimodal camera control framework for video generation. It generates videos that meet user expectations by receiving text and video as trajectory references, and images and videos as content references. We utilize LLM to extract input features, obtain camera motion trajectories through trajectory planning algorithms, and finally obtain complete videos through 3D reconstruction and diffusion models. To support the full-process training of OmniCam, we have constructed the OmniTr dataset - the first multimodal dataset specifically designed for camera control. Experimental results show that our model demonstrates excellent robustness when faced with different modal combination inputs, and can accurately generate camera trajectory videos that conform to user intentions.

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

# A APPENDIX

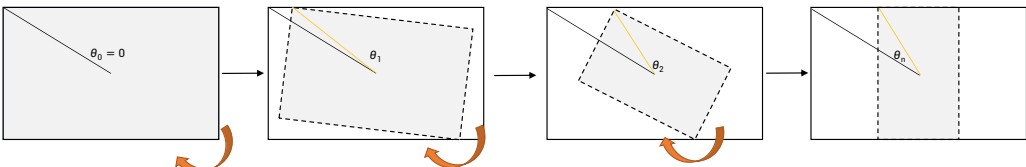

Figure 6: Visualization of Rotation Algorithm.

## A.1 DISCUSSION

**Why does VLM outperform MonST3R in Table 2?** Our test data includes complex camera trajectories and complex subject motions. Without deep semantic understanding, traditional extrinsic parameter estimation often confuses subject motion with camera motion. This problem is especially pronounced when the subject is large—for example, a drone following a moving car. The subject occupies a large portion of the scene, and traditional camera estimation methods may incorrectly conclude that the camera is stationary when, in fact, it is moving rapidly. This happens because traditional methods confuse the subject with the environment.

**Why not adopt Gaussian Splatting?** While existing reconstruction methods often rely on 3D Gaussian Splatting Kerbl et al. (2023) this technique involves a complex processing pipeline: it requires locating the Gaussian ellipsoid's center from point clouds, calculating the covariance matrix to construct the ellipsoid, adding opacity information, and finally rendering the video according to the target trajectory. Given the complexity of this process, we have chosen to directly utilize point clouds for monocular reconstruction and introduce a diffusion model to resolve the unknown regions in the rendering.

**Interactive video generation.** Currently, open-source solutions are unable to achieve true real-time performance—for instance, GameCrafter experiences action delays of up to 8 seconds in practice, whereas real-time interaction typically requires latency below 100 milliseconds. Achieving genuine real-time control and interaction has become a critical focus in the industry, which fundamentally differs from our goal of providing a multimodal interaction data engine.

## A.2 ROTATION ALGORITHM

As shown in Fig. 6, given a total rotation angle $\Theta$ (positive indicates counterclockwise rotation), the incremental rotation angle for each frame is $\Delta\theta = \frac{\Theta}{N}$, thus achieving a rotation angle of $\theta_k = k \times \Delta\theta$ at the $k$ frame. Assuming a rectangular image with dimensions $(w, h)$, after rotating around the center point $(\frac{w}{2}, \frac{h}{2})$, its projected width and height can be expressed as:

$$W(\theta) = w \cdot |cos\theta| + h \cdot |sin\theta|, \tag{5}$$

$$H(\theta) = w \cdot |sin\theta| + h \cdot |cos\theta|. \tag{6}$$

To ensure the rotated image maintains the original aspect ratio $(w \times h)$ without distortion, a scaling factor $s$ must be applied to the image, satisfying:

$$W(\theta) \times s \leq w, \quad H(\theta) \times s \leq h. \tag{7}$$

Therefore, we can derive:

$$s_w = \frac{w}{W(\theta)} = \frac{w}{w \cdot |cos\theta| + h \cdot |sin\theta|}, \tag{8}$$

$$s_h = \frac{h}{H(\theta)} = \frac{h}{w \cdot |sin\theta| + h \cdot |cos\theta|}, \tag{9}$$

$$s = min(s_w, s_h). \tag{10}$$

The visual frequency is adjusted according to the input timing, enabling smooth visual rotation. After generating all keyframe data $(\phi, \theta, r)$, the algorithm unifies them into a single trajectory sequence.

### A.3 METRICS

**CLIPSR** (CLIP-based Semantic Consistency Score) leverages the image encoding capabilities of the CLIP model to extract semantic features from each video frame and assesses the semantic coherence of the video by computing the similarity between frame features.

**NIQE** (Natural Image Quality Evaluator) is a no-reference image quality assessment method that evaluates image quality by extracting Natural Scene Statistics (NSS) features from the image and modeling them using a Multivariate Gaussian (MVG) model. It calculates the Mahalanobis distance between the test image features and the high-quality image statistical features. A lower score indicates a more natural image.

### A.4 MORE RELATED WORK: NOVEL VIEW SYNTHESIS

The process of transferring camera motion from video to static images is essentially a novel view synthesis problem. NeRF Mildenhall et al. (2021) utilizes neural networks to learn volumetric scene information from multiple viewpoints, predicting color and density for each spatial point through network parameter optimization to achieve realistic view synthesis. 3D Gaussian Splatting Kerbl et al. (2023) employs Structure-from-Motion (SfM) to obtain point clouds from multiple images, achieving near real-time 3D scene rendering by representing each point as a volumetric element (splat) with Gaussian distribution. While these methods typically require multiple images as input, users often can only provide a single image for creation. Consequently, researchers have begun exploring single-image novel view synthesis. Zero123 Liu et al. (2023) achieves new viewpoint image generation based on given camera poses by training diffusion models on synthetic datasets. TGS Zou et al. (2024) converts input images into Tri-plane feature representations and leverages 3D Gaussian Splatting for novel view rendering. However, these methods are limited to objects and consistently fail to generate 3D scenes. Recently, ZeroNVS Sargent et al. (2024) has achieved zero-shot novel view synthesis from a single input image through training on mixed datasets. Nevertheless, it struggles to synthesize consistent novel views and lacks precise pose control due to its treatment of camera pose conditions as high-level text embeddings. GenWarp Seo et al. (2024) combines text-image models with monocular depth estimation methods to generate new views, but its reconstruction remains unstable. ViewCrafter Yu et al. (2024) introduces a novel view synthesis framework that integrates video diffusion models with point cloud priors, achieving a balance between efficiency and fidelity.

### A.5 MORE DATASET DETAILS

The OmniTr dataset was initially created by using GPT to generate a large number of trajectory descriptions, which were then manually annotated with discrete motion representation information. Subsequently, these were converted into trajectory sequences in polar coordinate form through a trajectory planning algorithm, followed by the generation of corresponding videos using Viewcrafter. This data was later used for downstream training of OmniCam, essentially allowing OmniCam's downstream module to distill ViewCrafter's capabilities and further enhance them.

As shown in the Fig. 7, we conducted a visual analysis of the dataset to display data distribution, including word frequency distribution and motion feature statistics. The word cloud map shows that the keywords in the dataset mainly include time words (such as "second"), direction words (such as "right", "left", "upward", "downward"), speed words (such as "slow", "quick"), and motion words (such as "moves", "shifts", "rotors"). The first pie chart shows that single directional motion accounts for **27.5%**, compound directional motion accounts for **25.7%**, angular motion accounts for **40.1%**, and rotational motion accounts for the remaining **6.7%** in the dataset. The second pie chart shows the number of instructions for each group of texts. Most of the texts contain 2 to 4 groups of instructions.

This distribution reflects the comprehensive coverage of the dataset on different types of motion, especially the high proportion of angular motion and single-directional motion, which is consis-

tent with practical application scenarios. Meanwhile, the balanced proportion between combined directional movements and rotational motions ensures the diversity and equilibrium of the dataset.

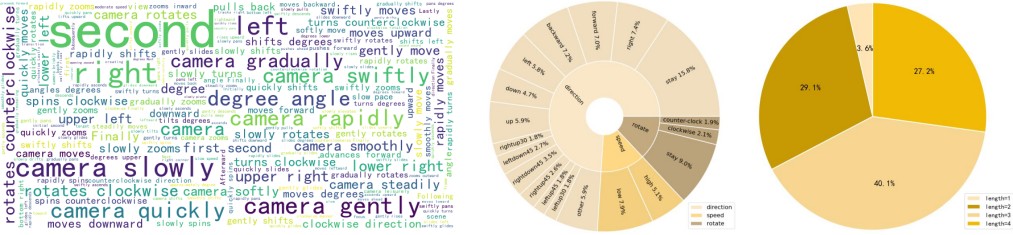

Figure 7: Visual analysis of the dataset.

## A.6 More Result

We provide more generation results of our model in Fig. 8. When confronted with different types of content references, our model can generate high-quality videos that follow the trajectory description.

## A.7 Comparison

Recent years have witnessed a flourishing development in the field of video generation, with numerous remarkable research outcomes emerging. As this field is still in its exploratory stage without unified evaluation standards or mature paradigms, it is challenging to conduct comprehensive and fair comparisons of various models solely through quantitative metrics. Here, we present a brief comparative analysis based on our practical usage experience to help understand their characteristics and limitations.

**CAT4D** Wu et al. (2024): A closed-source 4D reconstruction algorithm developed based on CAT3D. Due to its closed-source nature, specific details and performance are difficult to fully understand.

**DimensionX** Sun et al. (2024a): As a 4D reconstruction model, DimensionX currently only supports image input with limited control capabilities, allowing only basic control in left-right directions without adjusting degree, time, or speed parameters. Additionally, its generation speed is less than half that of OmniCam, and it does not support secondary editing of videos.

**DynamiCrafter** Xing et al. (2025): A tool focused on transforming static images or text descriptions into high-quality dynamic videos, with satisfactory dynamic content generation effects.

**ViewCrafter** Yu et al. (2024): This novel view generation model based on DynamiCrafter can generate perspective videos according to input trajectories; however, its capabilities in dynamic content generation are relatively limited.

**CogVideoX** Yang et al. (2024b): An excellent open-source video tool supporting text-to-video generation and image control, but lacking support for video input and trajectory migration.

**CameraCtrl** He et al. (2024): An excellent camera control tool trained on the RealEstate dataset, providing important inspiration for subsequent research. However, it lacks generalizability. Our experimental results show that camera trajectories selected from the RealEstate dataset perform well (the official inference code loads trajectories from this dataset). Since the RealEstate dataset contains many forward-moving camera effects, it performs well in forward and small-range backward movements. However, when we customize trajectories (such as "backward and right"), the spatial structure is compromised. Technically, it is less challenging for the camera to move to both sides while moving forward because the input image itself contains complete picture information, requiring less unknown information to be supplemented when moving forward and then right. Nevertheless, CameraCtrl is still excellent work, and we appreciate its contributions to the community.

**MotionMaster** Hu et al. (2024): This model can only provide content references through text and cannot control images. Its effects are demonstrated on the demo page.

*0 to 1 second, the camera steadily moves downward at a fast speed. 1 to 2 seconds, it slowly pans to the left. From 2 to 3 seconds, it rapidly pushes forward.*

*0-1 s, the camera steadily moves downward at a fast speed*    *1 to 2 seconds, it slowly pans to the left*    *2 to 3 seconds, it rapidly pushes forward.*

*From 0 to 2 seconds, the camera quickly moves forward. From 2 to 6 seconds, it quickly moves to up-right 30 degree.*

*From 0 to 2 seconds, the camera quickly moves forward*    *From 2 to 6 seconds, it quickly moves to up-right 30 degree*

*The camera first moves quickly upward. From 2 to 3 seconds, it quickly pans to the left.*

*The camera first moves quickly upward*    *From 2 to 3 seconds, it quickly pans to the left.*

*During the first 2 seconds, the camera slowly rises upward. Then, from 2 to 3 seconds, it quickly pushes downward.*

*0-2s, the camera slowly rises upward*    *2 to 3 seconds, it quickly pushes downward*

*0-3 seconds, the camera slowly shifts to the left.*

*0-3 seconds, the camera slowly shifts to the left*

*0-3 seconds, the camera moves rapidly and significantly horizontally downwards.*

*0-3 seconds, the camera moves rapidly and significantly horizontally downwards.*

*0-3 seconds, the camera slowly moves horizontally to the upper right at a 45-degree angle*

*0-3 seconds, the camera slowly moves horizontally to the upper right at a 45-degree angle*

*The camera first moves quickly upward. 1 to 3 seconds, it quickly moves to the left.*

*0-1 seconds, The camera first moves quickly upward*    *1 to 3 seconds, it quickly moves to the left.*

Figure 8: Some additional examples generated by OmniCam.

**CAMI2V** Zheng et al. (2024): This model is designed for image-to-video conversion. By inputting a reference image and camera trajectory parameters (or text), it outputs videos that follow specified camera movements but currently does not support complex operations such as rotation and camera zooming.

**CineMaster** Wang et al. (2025): Plans camera movement in 3D space and provides rich control capabilities, but the 3D operation workflow may still require some learning time for users unfamiliar with 3D modeling concepts. It also does not support video-guided trajectory generation.

**3DTrajMaster** Fu et al. (2025b): Primarily focuses on controlling object movement rather than camera motion, which limits its application in video generation.

**RealCaM** Li et al. (2025b): Performs poorly when input images are not realistic enough, limiting its practical application effects.

**Lucid-Dreamer** Liang et al. (2024): Its results show serious artifacts because it uses depth-based deformation to generate new perspectives, which is particularly problematic when processing wild images (with unknown camera intrinsics), leading to inaccurate new perspective generation. Additionally, it adopts a ready-made repair model [43] to optimize deformation results, but this often introduces inconsistencies between original and repaired content.

**ZeroNVS** Sargent et al. (2024): The quality of its generated new perspectives is relatively low with poor accuracy; the main reason is that ZeroNVS introduces camera pose conditions into the diffusion model through text embedding, which fails to provide precise control over new perspective generation, resulting in unsatisfactory results.

**MotionCtrl** Wang et al. (2024b): Can generate new perspectives with higher fidelity but performs inadequately in generating new perspectives precisely aligned with given camera conditions. This is because MotionCtrl also adopts high-level camera embeddings to control camera poses, resulting in lower accuracy in new perspective synthesis.

**DNGaussian** Li et al. (2024): A neural rendering method based on Gaussian distribution, aiming to generate high-quality scenes through probabilistic models. However, its results exhibit significant artifacts, indicating limited rendering capabilities in complex scenes.

**FSGS** Zhu et al. (2024): A fast scene generation method based on sparse Gaussian distribution, focusing on efficiently generating novel views. However, its results also show artifacts when viewed from novel perspectives that deviate from the ground truth training images, suggesting insufficient robustness in view extrapolation tasks.

**InstantSplat** Fan et al. (2024): A real-time scene generation method based on point clouds, utilizing DUSt3R for point cloud initialization, which better preserves details from the ground truth training images. However, due to its omission of the densification process, it fails to recover occlusion regions, resulting in incomplete performance in complex scenes.

Simultaneously, most works (such as CameraCtrl, RealCaM, and CAMI2V) are trained on the RealEstate dataset, lacking full-process training for trajectory control. This limitation affects their generalization ability and application in broader scenarios.

Furthermore, we also compared with some commercially available large models that have been launched:

**Hunyuan** Sun et al. (2024b): It has the powerful ability to generate videos from text, but it does not support providing trajectories through videos. Users need to input the corresponding camera trajectories in text form, and only simple zoom-in and zoom-out operations can be achieved.

**Tongyi** Cloud (2024): Has strong text-to-video generation capabilities but lacks spatial awareness, limiting its flexibility.

**Runway** Research (2023): An excellent commercial perspective control model with the best spatial consistency in benchmark tests. It supports image input and controls simple operations (such as"room in," "right," etc.) through buttons, allowing for the superposition of multiple directions but not supporting continuous multiple operations and time control.

Specifically, we compared the input modalities of multiple models. As shown in the Tab. 5, our model covers all input scenarios, facilitating various types of creation for users.

Table 5: Comparison of Content and Trajectory Reference Capabilities

| Model | Content Reference | | | Trajectory Reference | | |
|---|---|---|---|---|---|---|
| | **Text** | **Image** | **Video** | **Text** | **Video** | **Trajectory** |
| MotionMaster | ✓ | | | | ✓ | |
| OmniCam | ✓ | ✓ | ✓ | ✓ | ✓ | ✓ |
| CameraCtrl | ✓ | ✓ | | | ✓ | ✓ |
| CogVideoX | ✓ | | | ✓ | | |
| DimensionX | | ✓ | ✓ | | | ✓ |
| 3DTrajMaster | ✓ | | | | | ✓ |
| RealCam | ✓ | ✓ | | ✓ | | ✓ |
| CineMaster | ✓ | | | | | ✓ |
| CAMI2V | ✓ | ✓ | | | | ✓ |
| Runway | | ✓ | | ✓ | | |
| Hunyuan | ✓ | | | ✓ | | |
| Tongyi | ✓ | | | ✓ | | |

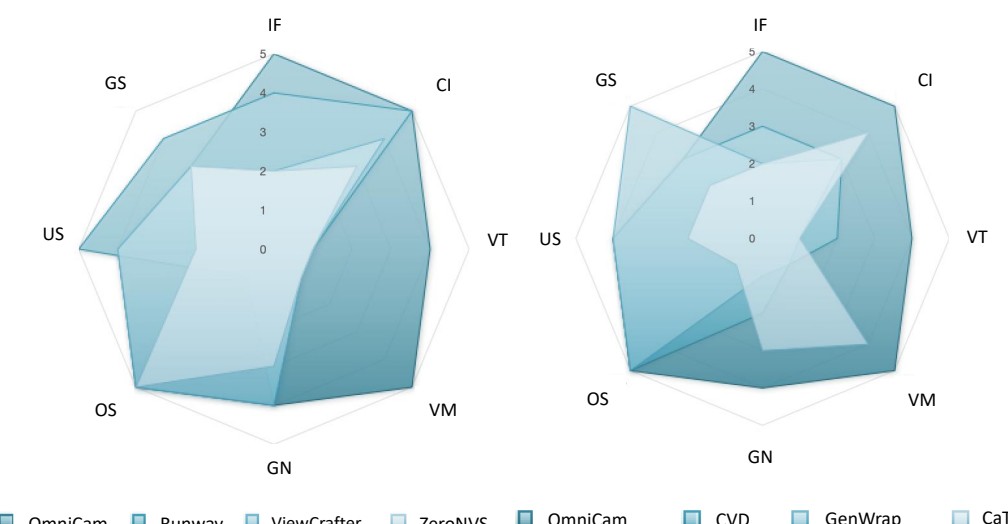

OmniCam    Runway    ViewCrafter    ZeroNVS    OmniCam    CVD    GenWrap    CaT4D

Figure 9: Radar chart comparing OmniCam with domain-specific models across eight dimensions: IF stands for Integration Flexibility, OS refers to Open Source, GS means Generation Speed, VM represents Video Manipulation, GN is Generalization, US indicates Usability, CI denotes Support Complex Instructions, and VT signifies Video Trajectory-Based.

## A.8 DETAILS CONCERNING TABLE 3

Since some methods Sargent et al. (2024); Liang et al. (2024) are only capable of processing square images, we crop the generated novel views from our method and MotionCtrl to ensure alignment when computing the quantitative metrics. On our demo page, we present extensive comparisons with related works. Currently, most reconstruction models focus primarily on object reconstruction, while research on scene reconstruction remains relatively limited. To comprehensively evaluate scene reconstruction capabilities, we selected several representative scene reconstruction models for comparative experiments. The experimental results demonstrate that DNGaussian exhibits significant artifacts in complex scenes, particularly in high dynamic range (HDR) or geometrically complex structures, where issues such as detail loss and edge blurring frequently occur. FSGS performs poorly in view extrapolation tasks; when the viewpoint deviates from the ground truth training images, the generated artifacts increase significantly, making it difficult to maintain geometric consistency and texture continuity. InstantSplat, which utilizes DUSt3R for point cloud initialization,

better preserves details from the ground truth training images. However, due to its omission of the densification process, it fails to effectively recover occluded regions, resulting in holes or distortions in complex scenes and compromising the overall visual quality. These limitations highlight the challenges in novel view synthesis tasks, particularly in achieving geometric consistency, texture continuity, and effective occlusion handling.

## A.9 MORE EXPERIMENTS

Table 6: Additional experiments compared with other methods.

| Model | RotErr↓ | TranErr↓ | PSNR↑ |
|---|---|---|---|
| CAT4D | 2.46 | 4.71 | 17.22 |
| SV4D | 3.39 | 16.52 | 8.48 |
| ViewCrafter | **1.09**(static only) | 2.86 | 22.04 |
| DynamiCrafter | 3.34 | 14.13 | **22.11** |
| TrajectoryCrafter | 1.23 | 2.64 | 21.77 |
| OmniCam | 1.15 | **2.63** | **22.11** |

To further demonstrate the effectiveness of our method, we conducted some capability comparisons with mainstream models such as CAT4D Wu et al. (2024), as shown in Table 6. In addition, due to the inability of ViewCrafter Yu et al. (2024) to handle dynamics, its evaluation does not include dynamic samples but only includes static samples, which makes it slightly outperform our method in terms of RotErr metrics. At the same time, OmniCam performs well on both dynamic samples and static samples.

