# OpenReview forum: "Omnicam: Unified Multimodal Video Generation via camera Control"
_ICLR.cc/2026/Conference — ICLR 2026 Conference Withdrawn Submission_

### Official Review · Reviewer_zgAa · 2025-10-31

**Soundness:** 2
**Presentation:** 3
**Contribution:** 2
**Rating:** 4
**Confidence:** 3

**Summary:**

The paper presents OmniCam, a unified multimodal camera control framework that enables diverse and precise visual effects by changing camera position and pose. Unlike previous methods that struggle with limited interaction modes and poor scene generalization, OmniCam integrates large language models and video diffusion models to generate spatially and temporally consistent videos. It supports flexible input combinations, allowing users to provide text or video as camera path guidance and image or video as content reference. To train the model, the authors introduce OmniTr, a large-scale dataset with over 10K scenes containing long video trajectories, videos, and detailed descriptions. Experiments show that OmniCam achieves state-of-the-art results in multimodal camera-controlled video generation.

**Strengths:**

Strengths
1. It introduces a multimodal approach that combines text, image, and video, which expands interaction possibilities and lowers user effort.
2. The proposed OmniTr dataset provides a valuable large-scale resource for training and evaluation.
5. The paper is clear in motivation and problem definition.

**Weaknesses:**

Weaknesses
1. The framework is complex and may be hard to reproduce or apply in real scenarios.
2. The model depends heavily on the large OmniTr dataset, which may affect generalization.
3. The ablation study is simple and does not clearly explain how each module contributes.

**Questions:**

Questions
1. Can the authors provide more details or guidelines to help reproduce the full framework?
2. In the ablation study, could the authors further explain how each module interacts and contributes to the final performance, possibly with visual or qualitative analysis?

---

### Official Review · Reviewer_sBFA · 2025-10-31

**Soundness:** 3
**Presentation:** 3
**Contribution:** 2
**Rating:** 4
**Confidence:** 4

**Summary:**

This paper introduces OmniCam, a framework for multimodal camera-controlled video generation. The system aims to unify text-, image-, and video-based inputs for guiding camera motion. It generates videos where users can specify both content (via image/video reference) and trajectory (via text or reference video).

The core pipeline consists of:
- Discrete Motion Representation (DMR): A symbolic encoding of camera movement (start time, end time, speed, direction, rotation).
- Trajectory Generation: Using a LoRA-tuned LLM to convert text (or a fine-tuned VLM for video) into DMR tokens.
- Trajectory Planning: Mapping DMR into 3D camera poses (ϕ, θ, r).
- Rendering + Diffusion: Projecting a reconstructed point cloud from the content reference to new camera viewpoints and filling missing regions via a video diffusion model (DiT).
- Dataset — OmniTr: A synthetic multimodal dataset with 10K+ videos, text descriptions, and camera trajectories, built largely from CO3D and ViewCrafter outputs.

The authors report strong results in rotation and translation accuracy, FID, and PSNR against models like CameraCtrl, MotionCtrl, and LucidDreamer, and demonstrate text- and video-based camera control on synthetic scenes.

**Strengths:**

- Unified pipeline: Handles combinations of (image | video) × (text | video) trajectory inputs — something prior works handled separately.
- New dataset (OmniTr): Synthetic but diverse; integrates text, video, and trajectory annotations for controllable camera motion.
- Strong empirical performance: Quantitative improvements over CameraCtrl and MotionCtrl in FID, PSNR, and trajectory accuracy.

**Weaknesses:**

- Paper writing is very unclear. The second paragraph in introduction makes no sense. Also in line 283, given one image, how is it possible to retrieve a camera pose using Dust3R?
- I am more confused about rendering on point cloud. So are you rendering on point map constructed by the first image. If so, how do you handle the back of the object?
- The paper titles as unified multimodal video diffusion via camera control and it is interesting to see this paper simplifies complicated 4D problem into first construct a dynamic 3D and then enable trajectory rendering. However, in the main text, I cannot find good examples of dynamic objects and camera control. While I find some examples in the supplementary materials (website), the camera motion is very small which is not convincing to me currently.

**Questions:**

- Does OmniCam generalize to real camera footage without retraining, or only synthetic datasets?
- This is more like editing task. Can we see original camera pose trajectory and you can edit some camera poses and show the results? I believe the text to discrete camera pose part, but I want to see more visualizations for the camera pose rendering from video diffusion.

---

### Official Review · Reviewer_Moje · 2025-10-31

**Soundness:** 2
**Presentation:** 1
**Contribution:** 3
**Rating:** 4
**Confidence:** 4

**Summary:**

OmniCam proposes a unified framework for fine-grained camera-controlled video generation across modalities. Its core is a discrete motion representation (start/end time, speed, direction, rotation) that translates either text or a reference video into a per-frame camera trajectory, while content comes from a single image or a video. A spherical trajectory planner turns the representation into extrinsics; point-cloud rendering provides geometry, and a DiT-based diffusion module inpaints unknown regions conditioned on the renders and CLIP features. To better match human perception, the system evaluates and adapts motion speed by bucketing scenes by scale. The paper also introduces OmniTr, a dataset with about 1K trajectories, 10K text descriptions, and 30K videos annotated at the sub-instruction level, and proposes trajectory-following metrics alongside standard video quality measures.

**Strengths:**

- **Simple and unified motion representation**
The discrete motion representation serves as a simple bridge that enables translation from multimodal conditions into control signals that guide the motion trajectory.

- **Text-to-trajectory at frame-level granularity**
Turning text prompts into multi-segment moves (with speed/angle/rotation) is practically valuable and provides a more user-friendly way to lower interaction costs.

- **Complete end-to-end path**
The system map text/video trajectories + image/video content all the way to output video, providing a more user-friendly and straightforward way for camera-controlled video generation.

**Weaknesses:**

- **Unclear writing, lack of technical details**
Lots of details are missing. For example, how does the condition (point cloud rendering, images, and videos) incorporate into the DiT model? What are the training specifics regarding section 4.1.1?

- **Dataset issues**
OmniTr Dataset appears to consist of synthetically generated videos (e.g., using ViewCrafter) and LLM-authored text, with some manual correction. Will this approach introduce any upper bounds or performance limitations due to the distribution gap?

- **Reliance on point-cloud reconstruction**
Heavy reliance on point-cloud reconstruction means failure cases can propagate to the diffusion stage. The ablation on sparse point clouds is useful, but wild real-world sequences with motion blur or rapid exposure changes are not convincingly stress-tested.

**Questions:**

See "Weaknesses"

---

### Official Review · Reviewer_8Wjo · 2025-11-01

**Soundness:** 3
**Presentation:** 3
**Contribution:** 2
**Rating:** 4
**Confidence:** 4

**Summary:**

This paper presents OmniCam, a unified framework for camera-controlled video generation that accepts multimodal inputs for both content (an image or video) and trajectory (text instructions or a reference video). The core technical approach is a multi-stage pipeline that decouples trajectory planning from content generation: first, an LLM parses the trajectory reference into a "Discrete Motion Representation" of commands; next, a planning algorithm converts this into a continuous 6DoF camera pose sequence. To synthesize the video, the system uses a "reconstruct-render-inpaint" process: it creates a 3D point cloud from the content, renders it from the new poses to create a draft with holes, and then uses a video diffusion model as a sophisticated in-painter to fill in these unknown regions. Finally, the paper introduces the OmniTr dataset, a large-scale synthetic dataset for training, and employs a reinforcement learning step to fine-tune the system, achieving state-of-the-art performance in trajectory accuracy.

**Strengths:**

- The paper's primary strength is its truly "unified" approach. The ability to mix and match input modalities for both content (Image/Video) and trajectory (Text/Video/Direct Params) is a step forward in usability and flexibility, addressing a clear need for creative and predictable control in video generation.
- The OmniTr dataset is a major contribution. A key bottleneck in camera control research has been the lack of large-scale, diverse, and well-annotated data. By generating 10,000 descriptions and 30,000 videos, the authors have provided a valuable resource. Furthermore, the introduction of new metrics (Mstarttime, Mspeed, Mdirection, etc.) to evaluate the trajectory parsing step is a great addition, allowing for fine-grained analysis of the system's "understanding" of the prompt.

**Weaknesses:**

- The paper is less of a technical contribution, but more of a complex, multi-stage system effort of at least four large, separately-trained components: an LLM, a SLAM/pose-extraction module, a monocular 3D reconstruction model, and a video diffusion model. This makes the system extremely complex to train, tune, and replicate. The "end-to-end" RL optimization is an attempt to couple two of these, but it's a post-hoc fix.
- The dataset creation process seems unclear to me. Is the OmniTr dataset consisted of real-world videos?

**Questions:**

N/A

---

### Note · Authors · 2025-11-12

**Comment:**

Thanks

**Withdrawal Confirmation:**

I have read and agree with the venue's withdrawal policy on behalf of myself and my co-authors.